# Clinicopathologic Analysis of Sinonasal Inverted Papilloma, with Focus on Human Papillomavirus Infection Status

**DOI:** 10.3390/diagnostics12020454

**Published:** 2022-02-10

**Authors:** Munechika Tsumura, Seiichiro Makihara, Asami Nishikori, Yuka Gion, Toshiaki Morito, Shotaro Miyamoto, Tomoyuki Naito, Kensuke Uraguchi, Aiko Oka, Tomoyasu Tachibana, Yorihisa Orita, Shin Kariya, Mitsuhiro Okano, Mizuo Ando, Yasuharu Sato

**Affiliations:** 1Department of Otolaryngology Head and Neck Surgery, Kagawa Rosai Hospital, Marugame 763-8502, Japan; mwfcf214@yahoo.co.jp (M.T.); shotarrow@s.okayama-u.ac.jp (S.M.); 2Department of Otolaryngology Head and Neck Surgery, Dentistry and Pharmaceutical Sciences, Graduate School of Medicine, Okayama University, Okayama 700-8558, Japan; pz6b2n7x@s.okayama-u.ac.jp (K.U.); skariya@cc.okayama-u.ac.jp (S.K.); ando-m@okayama-u.ac.jp (M.A.); 3Division of Pathophysiology, Graduate School of Health Sciences, Okayama University, Okayama 700-8558, Japan; asami.kei@s.okayama-u.ac.jp (A.N.); ghe421017@gmail.com (Y.G.); satou-y@okayama-u.ac.jp (Y.S.); 4Department of Medical Technology, Faculty of Health Sciences, Ehime Prefectural University of Health Sciences, Tobe 791-2101, Japan; 5Department of Pathology, Kagawa Rosai Hospital, Marugame 763-8502, Japan; morito_15j@yahoo.co.jp; 6Department of Otorhinolaryngology, Kagawa Prefectural Central Hospital, Takamatsu 760-8557, Japan; naitotomoyuki1@gmail.com; 7Department of Otorhinolaryngology, School of Medicine, International University of Health and Welfare, Narita 286-8520, Japan; aikooka@iuhw.ac.jp (A.O.); mokano@iuhw.ac.jp (M.O.); 8Department of Otolaryngology, Japanese Red Cross Society Himeji Hospital, Himeji 670-8540, Japan; tomoyasutachibana@hotmail.co.jp; 9Department of Otolaryngology Head and Neck Surgery, Graduate School of Medicine, Kumamoto University, Kumamoto 860-8556, Japan; y.orita@live.jp; 10Department of Pathology, Dentistry and Pharmaceutical Sciences, Graduate School of Medicine, Okayama University, Okayama 700-8558, Japan

**Keywords:** HPV infection, sinonasal inverted papilloma, diabetes mellitus, young adult, tumor stage

## Abstract

Sinonasal inverted papilloma (SNIP) can recur; however, the factors related to tumor recurrence remain unclear. This study aimed to analyze risk factors, including human papillomavirus (HPV) infection, as well as other factors associated with SNIP recurrence. Thirty-two patients who were diagnosed with SNIP and underwent surgery between 2010 and 2019 were enrolled: 24 men and 8 women, with a mean age of 59.2 years. The mean follow-up was 57.3 months. Demographics and information about history of smoking, diabetes mellitus (DM), hypertension, allergic rhinitis, alcohol consumption, tumor stage, surgical approach, and recurrence were reviewed retrospectively. Specimens were investigated using polymerase chain reaction to detect HPV DNA (high-risk subtypes: 16, 18, 31, 33, 35, 52b, and 58; low-risk subtypes: 6 and 11). Seven patients (21.9%) experienced recurrence. HPV DNA was detected in five (15.6%) patients (high-risk subtypes, *n* = 2; low-risk subtypes, *n* = 3). Patients with recurrence of SNIP had a higher proportion of young adults and displayed higher rates of HPV infection, DM, and advanced tumor stage than those without recurrence. HPV infection, young adulthood, DM, and advanced tumor stage could be associated with a high recurrence rate, which suggests that patients with these risk factors could require close follow-up after surgery.

## 1. Introduction

Sinonasal inverted papilloma (SNIP) is a benign neoplasm of the nasal cavity and paranasal sinuses and accounts for 0.5–4% of primary nasal tumors [1]. SNIP has a tendency to recur (12–20%), and malignant transformation has been found in 3–7% of cases [2]. Thus, many ear, nose, and throat surgeons are interested in understanding which factors are related to tumor recurrence.

The etiology of SNIP remains unknown. Certain hypotheses have been proposed, but causality has never been established for the suspected factors of smoking, allergy, and human papillomavirus (HPV) infection [3]. Several staging systems have been indicated to provide a recurrence rate-driven prognosis [4].

For more than 30 years, HPV has been suspected to play a major role in the pathophysiology of SNIP. Several studies have reported higher HPV detection rates in recurrent SNIP cases [5,6,7], but there is contradictory information about whether HPV-associated SNIP has a higher risk of recurrence [8,9].

Therefore, this study aimed to analyze risk factors that are associated with the recurrence of SNIP, containing HPV infection, as well as other risk factors such as age, sex, medical history, smoking, alcohol consumption, tumor stage, and surgical approach.

## 2. Materials and Methods

### 2.1. Patients

The study population consisted of 32 patients with SNIP who were treated at Kagawa Rosai Hospital between 2010 and 2019. All patients were pathologically diagnosed with SNIP (Figure 1). The mean duration of follow-up for the patients after SNIP detection was 57.3 months (range, 24–120 months).

We used opt-out to obtain consent for participation in this study.

Clinical characteristics were retrieved from patient medical records, which contained information on their medical history, including the presence of non-sinonasal papilloma, diabetes mellitus (DM), hypertension, and allergic rhinitis, as well as information on lifestyle characteristics (smoking and alcohol intake), occupational and industrial exposures such as welding fumes and organic solvents, primary symptoms, and complications after surgery. From the medical records, lifelong nonsmokers were defined as ‘nonsmokers’, and former smokers and current smokers as ‘smokers’. We also defined lifelong nondrinkers and social drinkers as ‘nondrinkers’, and those who regularly drink several times per week as ‘drinkers’.

All patients underwent preoperative computed tomography. The preoperative clinical stages of the initial surgeries were graded according to the Krouse staging system [10]. The tumor origin site was studied in relation to radiological features and intraoperative findings.

All surgeries were performed under general anesthesia using endoscopy, while external approaches, such as the Denker procedure or Caldwell–Luc operation, were added as needed for curative treatment. For complete surgical removal, accurate endoscopic identification and discrimination of the tumor from the surrounding normal mucosa were necessary. The tumor attachment site needed to be completely removed, and after it was removed, the part that was growing into the sinus was also extracted [11]. For the diagnosis of SNIP, histopathological examination was conducted for all patients. Patients with malignancies associated with tumors were excluded from this study. In each case, the specimen was analyzed and the following parameters were registered [12]: (i) enhanced hyperkeratosis or the presence of squamous hyperplasia, (ii) elevated mitotic index, (iii) lack of inflamed polyps, and (iv) greater number of aneuploid cells.

After surgical resection, the patients were followed up for at least two years to check for recurrence. Recurrence was defined as the presence of recurrent SNIP after initial curative treatment during the follow-up period. To rule out recurrent disease, patients were followed up after surgery, mainly by endoscopic examination.

### 2.2. HPV DNA Testing

HPV DNA was detected using consensus primer-mediated polymerase chain reaction (PCR) assays. Surgically removed tumor lesions were trimmed from formalin-fixed paraffin-embedded (FFPE) blocks of inverted papillomas, followed by deparaffinization, and total DNA was extracted using the QIAamp DNA Micro Kit (Qiagen, Valencia, CA, USA). A section with a thickness of 3 µm was prepared from the FFPE block, and the location of the lesion was confirmed by hematoxylin and eosin (HE) staining. Then, only the area with a high density of the tumor cells was dissected from the unstained slide, and DNA extraction was performed. PCR was performed using TaKaRa Taq^TM^ (Takara Bio, Tokyo, Japan) and primers that detected HPV DNA, as reported previously [13,14,15]. The primers designed to target the E6 and E7 genes of HPV were a forward primer for low-risk HPV: 5′-TGCTAATTCGGTGCTACCTG-3′; a forward primer for high-risk HPV: 5′-TGTCAAAAACCGTTGTGTCC-3′; and reverse primer: 5′-GAGCTGTCGCTTAATTGCTC-3′. The reverse primer was fluorescently labeled, and the PCR product was analyzed using ABI PRISM 310 (Applied Biosystems, Foster City, CA, USA) and GeneMapper v3.7 (Applied Biosystems, Waltham, MA, USA). We defined a specimen as HPV-positive when a peak corresponding to a band size of 228 to 268 bp was detected (Figure 2). Since the positive and negative controls for the HPV PCR assay have already been reported in the past, HPV PCR was determined based on them [15]. The seven HPV variants in Group 1 were considered as high-risk types (HPV 16, 18, 31, 33, 35, 52b, and 58), and the two HPV variants in Group 2 were considered as low-risk types (HPV 6 and 11) [13,14].

We also performed PCR on internal control genes to assess the validity of the HPV DNA amplification results. PCR was performed using the primers for β-actin and GAPDH [16,17], and amplification products were confirmed by electrophoresis on a 3% agarose gel.

### 2.3. Statistical Analysis

Differences in baseline characteristics were assessed using the chi-square test or Fisher’s exact test, as appropriate. All statistical analyses were conducted using the statistical software ‘EZR’ (Easy R) [18]. Values with *p* < 0.05 were accepted as significant, while those with *p* < 0.1 were considered to indicate a tendency.

## 3. Results

### 3.1. Overall Outcomes

The 32 unoperated patients with SNIP included 24 men and 8 women, with a mean age of 59.2 years (range, 27–84 years) at the time of papilloma detection. In all cases, gross total resection was performed. Seven patients (21.9%) experienced a recurrence. Nineteen (59.4%) were smokers and 9 (28.1%) drinkers. The number of patients with non-sinonasal papilloma, allergic rhinitis, DM, or hypertension was 0 (0%), 10 (31.3%), 5 (15.6%), and 11 (34.4%), respectively. There were no patients with occupational and industrial exposures, such as welding fumes or organic solvents. There were 27 patients with nasal obstruction, 4 with epistaxis, and 1 with postnasal drip. The number of tumor attachment sites varied between groups. There were 2 patients (6.3%) with Krouse stage T1, 9 (28.1%) with Krouse stage T2, 21 (65.6%) with Krouse stage T3, and 0 (0%) with Krouse stage T4. An endoscopic approach was performed in 29 patients, and combined approaches (both endoscopic and external approaches) in 3 patients. There were two patients with epistaxis and two with numbness of the cheek as complications. We could control epistaxis in both patients using bipolar cautery. The numbness of the cheek in two patients was resolved a few months after surgery. No major complications were observed. There were 15 patients (46.9%) with enhanced hyperkeratosis or presence of squamous hyperplasia, 0 (0%) with an elevated mitotic index, 0 (0%) with a lack of inflamed polyps, and 0 (0%) with a greater number of aneuploid cells. HPV DNA was detected in 5 (15.6%) of the 32 patients with SNIP. Of these, two were positive for high-risk HPV subtypes, and three were positive for low-risk HPV subtypes.

### 3.2. Association between Recurrence and Clinical Characteristics

Younger adults (<40 years) showed significant associations with recurrence compared to older adults (≥40 years; *p* = 0.025) (Table 1). Patients with DM showed significant associations with recurrence compared to those without DM (*p* < 0.01). Patients with Krouse stage T3 showed significant associations with recurrence compared to those with Krouse stages T1 or T2 (*p* = 0.030). Additionally, patients who had HPV high- or low-risk variants showed significant associations with recurrence, compared to those who were negative for HPV variants (*p* = 0.025).

### 3.3. Association between HPV Infection and Clinical Characteristics

Younger patients (<40 years) showed significantly stronger associations with HPV infection (positive for high- or low-risk HPV DNA) than older patients did (≥40 years; *p* < 0.01) (Table 2).

## 4. Discussion

The results of our research on the relationship between clinical or demographic data and recurrence of SNIP showed that younger age (<40 years), presence of DM, advanced tumor stage, and HPV DNA were associated with higher recurrence rates. Pähler et al. also reported that patients who presented with recurrent papilloma infection were significantly younger (48.7 years old on average) at the time of initial diagnosis than those (60.2 years old on average) with non-recurrent tumors (*p* = 0.0194), and multivariate logistic regression revealed that a younger age at initial diagnosis was the strongest risk factor for neoplasm recurrence [19]. In contrast, several reports found no relationship between recurrence and younger age [4,20,21]. Therefore, a detailed investigation with a large sample size is needed.

Patients with impaired immune responses have a greater tendency to develop HPV-associated disorders [22,23,24]. In patients with DM, a common chronic disease in Japan, the proliferation of macrophages and T cells is altered, and the function of B and NK cells is impaired, which results in abnormal innate and adaptive immunity [25]. Several biological mechanisms may possibly increase the incidence of HPV-related anogenital precancer and cancer occurrence in women with diabetes [26]. As one such mechanism, hyperglycemia in diabetes is related to cell-mediated immune deficiencies and an increased vulnerability to viral infections and, which may endanger clearance of HPV infections and thus boost progression to precancer and cancer development [26,27,28]. Moon et al. reported no relationship between recurrence and DM in 132 cases of SNIP [20]. The present study is the first to report the relationship between DM and SNIP recurrence, in addition to DM and HPV infections among patients with SNIP, partially because a limited number of studies examined SNIP and HPV infections.

Krouse [10], Han et al. [29], Cannady et al. [30], and others have proposed several classification staging systems for SNIP. The most widely used classification system is that of Krouse, which emphasizes that tumor extension over the medial maxillary sinus or to the frontal or sphenoid sinus is an important prognostic factor [20,31]. Of the recurrences reported by Gras-Cabrerizo et al. [31] according to the Krouse system: 0% occurred in the T1 stage, 16% in the T2 stage, 25% in the T3 stage, and 60% in the T4 stage (*p* = 0.05). Furthermore, Moon et al. reported that the Krouse stage T4 group had more frequent recurrences than the T1, T2, and T3 groups did [20]. This study found that patients with Krouse stage T3 showed significant associations with recurrence, compared to those with T1 or T2.

HPV, an epitheliotropic DNA virus, can infect the epidermis or mucosa in humans. During infection, viral DNA sequences of HPV are incorporated into cellular DNA and play a key role in the promotion of neoplasm growth and malignant transformation of SNIP [32]. The detection method and detection rate of HPV-PCR are important points, and there have been several studies that employ HPV-PCR in FFPE samples [33,34,35]. The internal control gene amplification was performed to assess the validity of the results obtained in this study. β-actin and GAPDH were used as the internal controls [16,17]. β-actin (258 bp) was amplified in 31 of 32 cases, and GAPDH (226 bp) was amplified in all cases (Appendix A). These results suggest that HPV-PCR approach is unlikely to result in false negatives due to DNA fragmentation because of formalin fixation. Consistent with detections from previous reports, those with malignant and benign clinical courses could be separated based on high- and low-risk types of HPV, respectively. HPV-6 and -11 are considered as low-grade risk types, while HPV-16 and -18 are considered as high-grade risk types [36]. The detection rate of HPV was increased in SNIP with carcinoma and high-grade dysplasia, as compared to that in SNIP with mild dysplasia or no dysplasia [37]. Meta-analyses showed a significant relationship between HPV infection and malignant transformation of SNIP [38,39]. Additionally, several studies reported that patients who were SNIP positive for HPV infection showed higher levels of recurrence than those who were SNIP-negative for HPV infection [5,6,7]. In contrast, a study of 57 patients reported HPV DNA in seven SNIP areas, all of which were grades II and I (benign) SNIP cases. All other cases were grade III or grade IV (carcinoma arising from SNIP) and negative for HPV DNA. High-risk HPV subtype DNA was found in five of seven cases, suggesting that infection, especially with high-risk HPV subtypes, was an early and stimulating event in tumorigenesis [40]. Taken together, these studies suggest that HPV plays a key role in not only malignant transformation, but also early pathological development and recurrence [36].

Smoking is considered the most momentous risk factor for the development and recurrence of tumors occurring in the head, neck, and uterine cervix regions [36,41]. As one of the environmental risk factors, smoking has been associated with recurrence of SNIP. In a study of 132 patients, SNIP had recurrence in 21 of 132 (15.9%) patients. While 11 of the 39 smokers (28.2%) had recurrence of the disease, only 10 of the 93 nonsmokers (10.7%) had recurrence, which showed a significantly higher rate of recurrence among smokers (*p* = 0.012) [20]. This study found no significant difference between nonsmokers and smokers in the incidence of SNIP recurrence.

A systematic review and meta-analysis indicated that an endoscopic approach was an approving treatment option for SNIP and confirmed a global recommendation that it is the world standard in the treatment of such nose lesions, discovering a lower recurrence rate than that in external approaches. However, the recurrence rate did not show a significant difference between the endoscopic and combined approach groups [42]. Even when an external approach is necessary, the combination of an endoscopic approach is important for suppressing recurrence, as was observed in this study.

Despite being performed in a single institution, this study had limitations. First, the low positivity rate of HPV DNA made it difficult to evaluate the influence of HPV infection on the SNIP recurrence. Only 5 of 32 patients (15.6%) were positive for HPV DNA in the present study. Three recent PCR-based studies detected HPV in 11 of 90 patients (12.2%) [37], 2 of 19 patients (10.5%) [43], and 8 of 54 patients (14.8%) [4]. The low prevalence of HPV reflects the possibility that HPV infection is not the main causal factor for the pathogenesis of SNIP. Second, there were few patients with the T1 stage according to the Krouse system, and no patient with T4 stage. Furthermore, the number of young patients recruited in this study was small. A multicenter study with a large sample size is needed to establish the relationship between SNIP recurrence and HPV infection and identify additional risk factors associated with recurrence.

## 5. Conclusions

This study examined the risk factors for SNIP recurrence after surgical resection. HPV infection, young adulthood, DM, and advanced tumor stage were associated with a high recurrence rate. These results suggest that patients with these risk factors require information about risk factors and close follow-up after surgery. In the future, a multicenter study with a large sample size will be conducted to establish the relationship between SNIP recurrence and HPV infection, and the relationship between recurrence and other risk factors.

## Figures and Tables

**Figure 1 diagnostics-12-00454-f001:**
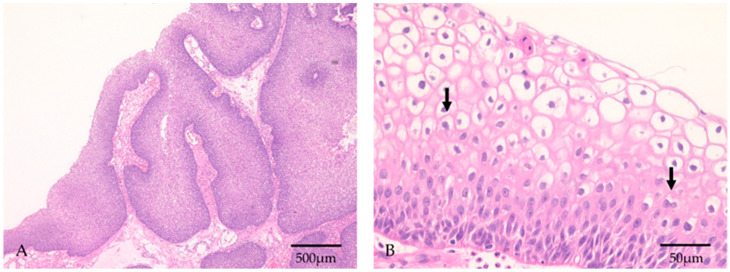
Histopathological features of HPV-positive SNIP. (**A**) The squamous epithelium showed an inverted growth pattern. Objective magnification: 4×. (**B**) Koilocytosis was observed, which has a perinuclear halo. Some koilocytes showed nuclear atypia and binuclear cells (arrows). Objective magnification: 40×.

**Figure 2 diagnostics-12-00454-f002:**
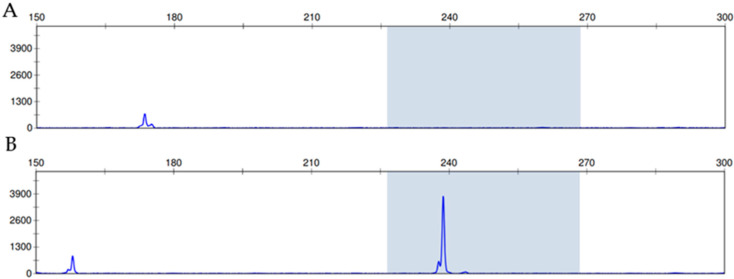
Criteria for fragment analysis. (**A**) Negative case, no peak at the expected fragment size of 228–268 bp (**B**) Positive case, a peak that is detected from the PCR product and represents a fragment size of 228–268 bp.

**Table 1 diagnostics-12-00454-t001:** Association between Recurrence and Clinical Characteristics in Patients with Sinonasal Inverted Papilloma.

		Recurrence		
		(+)	(−)	*p* Value
Sex	Male	6	18	
	Female	1	7	0.46
Age	≥40	4	23	
	<40	3	2	0.02
Smoker	Yes	5	14	
	No	2	11	0.46
Drinker	Yes	3	6	
	No	4	19	0.33
History of allergic rhinitis	Yes	3	7	
	No	4	18	0.45
History of DM	Yes	4	1	
	No	3	24	<0.01
History of hypertension	Yes	3	8	
	No	4	17	0.59
Staging	T1 or T2	0	11	
	T3	7	14	0.03
Surgical methods	Endoscopic excision	6	23	
	Endoscopic excision combined with external approach	1	2	0.61
EH or presence of SH	(+)	3	12	
	(−)	4	13	0.81
HPV high or low risk (PCR)	(+)	3	2	
	(−)	4	23	0.02

DM = diabetes mellitus; EH = enhanced hyperkeratosis; SH = squamous hyperplasia. (+) = recurrence+; (−) = recurrence−.

**Table 2 diagnostics-12-00454-t002:** Association Between HPV Infection and Clinical Characteristics in Patients with Sinonasal Inverted Papilloma.

		HPV High or Low Risk (PCR)		
		(+)	(−)	*p* Value
Sex	Male	5	19	
	Female	0	8	0.16
Age	≥40	2	25	
	<40	3	2	<0.01
Smoker	Yes	4	15	
	No	1	12	0.31
Drinker	Yes	1	8	
	No	4	19	0.66
History of allergic rhinitis	Yes	2	8	
	No	3	19	0.65
History of DM	Yes	2	3	
	No	3	24	0.1
History of hypertension	Yes	2	9	
	No	3	18	0.77
Staging	T1 or T2	0	11	
	T3	5	16	0.08
Surgical methods	Endoscopic excision	5	24	
	Endoscopic excision combined with external approach	0	3	0.43
EH or presence of SH	(+)	3	12	
	(−)	2	15	0.52
Recurrence	Yes	3	4	
	No	2	23	0.02

DM, diabetes mellitus; EH, enhanced hyperkeratosis; SH, squamous hyperplasia. (+) = detected; (−) = not detected.

## Data Availability

Not applicable.

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
