# Peer review of "Clinicopathologic Analysis of Sinonasal Inverted Papilloma, with Focus on Human Papillomavirus Infection Status"

_diagnostics, 2022, doi:10.3390/diagnostics12020454_

Round 1
Reviewer 1 Report
The manuscript is sound
Author Response
My paper has been edited by Editage, the English language editing service. I will attach the editing certificate.
Reviewer 2 Report
The authors conducted a study to analyse risk factors for recurrence of sinonasal inverted papilloma (SNIP). Concretely, the authors studied the potential relationship between recurrence of SNIP and human papillomavirus (HPV) infection, lifestyle and demographic characteristics, among other factors.
General comments
The article is clear and interesting to read. However, there are a few concerns that should be addressed prior to this paper is considered for publication:
- Abstract, last sentence: Taking into account the size of the sample (n=32), I would advise authors to avoid using sentences like “HPV infection, young adulthood, DM, and advanced tumor stage are associated with a high recurrence rate, which suggests that patients with these risk factors require close follow-up after surgery”. It should be better: HPV infection, young adulthood, DM, and advanced tumor stage could be associated with a high recurrence rate, which suggests that patients with these risk factors could require close follow-up after surgery. Don´t forget that only 7 patients had recurrence, so estimates have huge error margins. Please, review the rest of the manuscript.
- Introduction, 3rd paragraph: Please, include some references which corroborate that there is contradictory information about whether HPV-associated SNIP has a higher risk of recurrence.
- Some sentences of the introduction and the discussion are very similar to their original. Authors must re-write them. In general, there should not be consecutive 6 words copied. You can use duplication check websites to inspect your manuscript before you send it back.
- The references list must be reviewed (see https://www.mdpi.com/journal/diagnostics/instructions#references). For example, abbreviated journal names (review references #5, 6, 7…).
Other comments:
- Caption of the figure 2. The caption of the figure 2 should appear just below the figure.
Author Response
Response to Reviewer 2 Comments
General comments
The article is clear and interesting to read. However, there are a few concerns that should be addressed prior to this paper is considered for publication:
- Abstract, last sentence: Taking into account the size of the sample (n=32), I would advise authors to avoid using sentences like “HPV infection, young adulthood, DM, and advanced tumor stage are associated with a high recurrence rate, which suggests that patients with these risk factors require close follow-up after surgery”. It should be better: HPV infection, young adulthood, DM, and advanced tumor stage could be associated with a high recurrence rate, which suggests that patients with these risk factors could require close follow-up after surgery. Don´t forget that only 7 patients had recurrence, so estimates have huge error margins. Please, review the rest of the manuscript.
Thank you for your comment and suggestion. We have revised the manuscript accordingly and hope that our revisions address all your concerns.
We have revised the abstract as follows:
“HPV infection, young adulthood, DM, and advanced tumor stage could be associated with a high recurrence rate, which suggests that patients with these risk factors could require close follow-up after surgery.”
- Introduction, 3rd paragraph: Please, include some references which corroborate that there is contradictory information about whether HPV-associated SNIP has a higher risk of recurrence.
Thank you for your suggestion; we have added two references (#8, 9).
- Some sentences of the introduction and the discussion are very similar to their original. Authors must re-write them. In general, there should not be consecutive 6 words copied. You can use duplication check websites to inspect your manuscript before you send it back.
Thank you for your suggestion. We have modified the sentences using duplication check service. We have highlighted the changes made in the manuscript in yellow.
- The references list must be reviewed (see https://www.mdpi.com/journal/diagnostics/instructions#references). For example, abbreviated journal names (review references #5, 6, 7…).
Thank you for your suggestion and we apologize for the formatting error. We have ensured that all citations are correctly formatted in accordance with the Journal’s guidelines.
Other comments:
- Caption of the figure 2. The caption of the figure 2 should appear just below the figure.
Thank you for your suggestion, we revised Figure 2’s caption accordingly.